# Are Transformers More Robust Than CNNs?

**Yutong Bai[1]**    **Jieru Mei[1]**    **Alan Yuille[1]**    **Cihang Xie[2]**

[1]Johns Hopkins University          [2] University of California, Santa Cruz

{ytongbai, meijieru, alan.l.yuille, cihangxie306}@gmail.com

## Abstract

Transformer emerges as a powerful tool for visual recognition. In addition to demonstrating competitive performance on a broad range of visual benchmarks, recent works also argue that Transformers are much more robust than Convolutions Neural Networks (CNNs). Nonetheless, surprisingly, we find these conclusions are drawn from unfair experimental settings, where Transformers and CNNs are compared at different scales and are applied with distinct training frameworks. In this paper, we aim to provide the first **fair & in-depth** comparisons between Transformers and CNNs, focusing on robustness evaluations.

With our unified training setup, we first challenge the previous belief that Transformers outshine CNNs when measuring *adversarial robustness*. More surprisingly, we find CNNs can easily be as robust as Transformers on defending against adversarial attacks, if they properly adopt Transformers' training recipes. While regarding *generalization on out-of-distribution samples*, we show pre-training on (external) large-scale datasets is not a fundamental request for enabling Transformers to achieve better performance than CNNs. Moreover, our ablations suggest such stronger generalization is largely benefited by the Transformer's self-attention-like architectures per se, rather than by other training setups. We hope this work can help the community better understand and benchmark the robustness of Transformers and CNNs. The code and models are publicly available at https://github.com/ytongbai/ViTs-vs-CNNs.

## 1   Introduction

Convolutional Neural Networks (CNNs) have been the widely-used architecture for visual recognition in recent years [22, 38, 40, 16, 21]. It is commonly believed the key to such success is the usage of the convolutional operation, as it introduces several useful inductive biases (*e.g.*, translation equivalence) to models for benefiting object recognition. Interestingly, recent works alternatively suggest that it is also possible to build successful recognition models without convolutions [34, 60, 3]. The most representative work in this direction is Vision Transformer (ViT) [12], which applies the pure self-attention-based architecture to sequences of images patches and attains competitive performance on the challenging ImageNet classification task [35] compared to CNNs. Later works [26, 47] further expand Transformers with compelling performance on other visual benchmarks, including COCO detection and instance segmentation [23], ADE20K semantic segmentation [61].

The dominion of CNNs on visual recognition is further challenged by the recent findings that Transformers appear to be *much more robust* than CNNs. For example, Shao *et al*. [37] observe that the usage of convolutions may introduce a negative effect on models' adversarial robustness, while migrating to Transformer-like architectures (*e.g.*, the Conv-Transformer hybrid model or the pure Transformer) can help secure models' adversarial robustness. Similarly, Bhojanapalli *et al*. [4] report that, if pre-trained on sufficiently large datasets, Transformers exhibit considerably stronger robustness than CNNs on a spectrum of out-of-distribution tests (*e.g.*, common image corruptions [17], texture-shape cue conflicting stimuli [13]).

35th Conference on Neural Information Processing Systems (NeurIPS 2021).

Though both [4] and [37] claim that Transformers are preferable to CNNs in terms of robustness, we find that such conclusion cannot be strongly drawn based on their existing experiments. Firstly, Transformers and CNNs are not compared at the same model scale, *e.g.*, a small CNN, ResNet-50 (~25 million parameters), by default is compared to a much larger Transformer, ViT-B (~86 million parameters), for these robustness evaluations. Secondly, the training frameworks applied to Transformers and CNNs are distinct from each other (*e.g.*, training datasets, number of epochs, and augmentation strategies are all different), while little efforts are devoted on ablating the corresponding effects. In a nutshell, due to these inconsistent and unfair experiment settings, *it remains an open question whether Transformers are truly more robust than CNNs*.

To answer it, in this paper, we aim to provide the first benchmark to fairly compare Transformers to CNNs in robustness evaluations. We particularly focus on the comparisons between Small Data-efficient image Transformer (DeiT-S) [43] and ResNet-50 [16], as they have similar model capacity (*i.e.*, ~22 million parameters *vs.* ~25 million parameters) and achieve similar performance on ImageNet (*i.e.*, 76.8% top-1 accuracy *vs.* 76.9% top-1 accuracy[1]). Our evaluation suite accesses model robustness in two ways: 1) adversarial robustness, where the attackers can actively and aggressively manipulate inputs to approximate the worst-case scenario; 2) generalization on out-of-distribution samples, including common image corruptions (ImageNet-C [17]), texture-shape cue conflicting stimuli (Stylized-ImageNet [13]) and natural adversarial examples (ImageNet-A [19]).

With this unified training setup, we present a completely different picture from previous ones [37, 4]. Regarding adversarial robustness, we find that Transformers actually are no more robust than CNNs— if CNNs are allowed to properly adopt Transformers' training recipes, then these two types of models will attain similar robustness on defending against both perturbation-based adversarial attacks and patch-based adversarial attacks. While for generalization on out-of-distribution samples, we find Transformers can still substantially outperform CNNs even without the needs of pre-training on sufficiently large (external) datasets. Additionally, our ablations show that adopting Transformer's self-attention-like architecture is the key for achieving strong robustness on these out-of-distribution samples, while tuning other training setups will only yield subtle effects here. We hope this work can serve as a useful benchmark for future explorations on robustness, using different network architectures, like CNNs, Transformers, and beyond [42, 24].

## 2 Related Works

**Vision Transformer.** Transformers, invented by Vaswani *et al.* in 2017 [46], have largely advanced the field of natural language processing (NLP). With the introduction of self-attention module, Transformer can effectively capture the non-local relationships between all input sequence elements, achieving the state-of-the-art performance on numerous NLP tasks [54, 10, 5, 11, 31, 32].

The success of Transformer on NLP also starts to get witnessed in computer vision. The pioneering work, ViT [12], demonstrates that the pure Transformer architectures are able to achieve exciting results on several visual benchmarks, especially when extremely large datasets (*e.g.*, JFT-300M [39]) are available for pre-training. This work is then subsequently improved by carefully curating the training pipeline and the distillation strategy to Transformers [43], enhancing the Transformers' tokenization module [55], building multi-resolution feature maps on Transformers [26, 47], designing parameter-efficient Transformers for scaling [57, 45, 52], *etc*. In this work, rather than focusing on furthering Transformers on standard visual benchmarks, we aim to provide a fair and comprehensive study of their performance when testing out of the box.

**Robustness Evaluations.** Conventional learning paradigm assumes training data and testing data are drawn from the same distribution. This assumption generally does not hold, especially in the real-world case where the underlying distribution is too complicated to be covered in a (limited-sized) dataset. To properly access model performance in the wild, a set of robustness generalization benchmarks have been built, *e.g.*, ImageNet-C [17], Stylized-ImageNet [13], ImageNet-A [19], *etc*. Another standard surrogate for testing model robustness is via adversarial attacks, where the attackers deliberately add small perturbations or patches to input images, for approximating the worst-case evaluation scenario [41, 14]. In this work, both robustness generalization and adversarial robustness are considered in our robustness evaluation suite.

---

[1]Here we apply the general setup in [44] for the ImageNet training. We follow the popular ResNet's standard to train both models for 100 epochs. Please refer to Section 3.1 for more training details.

Concurrent to ours, both Bhojanapalli *et al.* [4] and Shao *et al.* [37] conduct robustness comparisons between Transformers and CNNs. Nonetheless, we find their experimental settings are unfair, *e.g.*, models are compared at different capacity [4, 37] or are trained under distinct frameworks [37]. In this work, our comparison carefully align the model capacity and the training setups, which draws completely different conclusions from the previous ones.

## 3 Settings

### 3.1 Training CNNs and Transformers

**Convolutional Neural Networks.** ResNet [16] is a milestone architecture in the history of CNN. We choose its most popular instantiation, *ResNet-50* (with ~25 million parameters), as the default CNN architecture. To train CNNs on ImageNet, we follow the standard recipe of [15, 33]. Specifically, we train all CNNs for a total of 100 epochs, using momentum-SGD optimizer; we set the initial learning rate to 0.1, and decrease the learning rate by $10\times$ at the 30-th, 60-th, and 90-th epoch; no regularization except weight decay is applied.

**Vision Transformer.** ViT [12] successfully introduces Transformers from natural language processing to computer vision, achieving excellent performance on several visual benchmarks compared to CNNs. In this paper, we follow the training recipe of DeiT [43], which successfully trains ViT on ImageNet without any external data, and set *DeiT-S* (with ~22 million parameters) as the default Transformer architecture. Specifically, we train all Transformers using AdamW optimizer [27]; we set the initial learning rate to 5e-4, and apply the cosine learning rate scheduler to decrease it; besides weight decay, we additionally adopt three data augmentation strategies (*i.e.*, RandAug [9], MixUp [59] and CutMix [56]) to regularize training (otherwise DeiT-S will attain significantly lower ImageNet accuracy due to overfitting [6]).

Note that different from the standard recipe of DeiT (which applies 300 training epochs by default), we hereby train Transformers only for a total of 100 epochs, *i.e.*, same as the setup in ResNet. We also remove {Erasing, Stochastic Depth, Repeated Augmentation}, which were applied in the original DeiT framework, in this basic 100 epoch schedule, for preventing over-regularization in training. Such trained DeiT-S yields 76.8% top-1 ImageNet accuracy, which is similar to the ResNet-50's performance (76.9% top-1 ImageNet accuracy).

### 3.2 Robustness Evaluations

Our experiments mainly consider two types of robustness here, *i.e.*, robustness on adversarial examples and robustness on out-of-distribution samples.

**Adversarial Examples**, which are crafted by adding human-imperceptible perturbations or small-sized patches to images, can lead deep neural networks to make wrong predictions. In addition to the very popular PGD attack [28], our robustness evaluation suite also contains: A) AutoAttack [8], which is an ensemble of diverse attacks (*i.e.*, two variants of PGD attack, FAB attack [7] and Square Attack [1]) and is parameter-free; and B) Texture Patch Attack (TPA) [53], which uses a predefined texture dictionary of patches to fool deep neural networks.

Recently, several benchmarks of **out-of-distribution samples** have been proposed to evaluate how deep neural networks perform when testing out of the box. Particularly, our robustness evaluation suite contains three such benchmarks: A) ImageNet-A [19], which are real-world images but are collected from challenging recognition scenarios (*e.g.*, occlusion, fog scene); B) ImageNet-C [17], which is designed for measuring model robustness against 75 distinct common image corruptions; and C) Stylized-ImageNet [13], which creates texture-shape cue conflicting stimuli by removing local texture cues from images while retaining their global shape information.

## 4 Adversarial Robustness

In this section, we investigate the robustness of Transformers and CNNs on defending against adversarial attacks, using ImageNet validation set (with 50,000 images). We consider both perturbation-based attacks (*i.e.*, PGD and AutoAttack) and patch-based attacks (*i.e.*, TPA) for robustness evaluations.

## 4.1 Robustness to Perturbation-Based Attacks

Following [37], we first report the robustness of ResNet-50 and DeiT-S on defending against AutoAttack. We verify that, when applying with a small perturbation radius $\epsilon = 0.001$, DeiT-S indeed achieves higher robustness than ResNet-50, *i.e.*, 22.1% *vs.* 17.8% as shown in Table 1.

However, when increasing the perturbation radius to 4/255, a more challenging but standard case studied in previous works [36, 48, 49], *both models will be circumvented completely*, *i.e.*, 0% robustness on defending against AutoAttack. This is mainly due to that both models are not adversarially trained [14, 28], which is an effective way to secure model robustness against adversarial attacks, and we will study it next.

Table 1: Performance of ResNet-50 and DeiT-S on defending against AutoAttack, using ImageNet validation set. We note both models are completely broken when setting perturbation radius to 4/255.

|  | Clean | Perturbation Radius | |
|---|---|---|---|
|  |  | 0.001 | 4/255 |
| ResNet-50 | 76.9 | 17.8 | 0.0 |
| DeiT-S | 76.8 | 22.1 | 0.0 |

### 4.1.1 Adversarial Training

Adversarial training [14, 28], which trains models with adversarial examples that are generated on-the-fly, aims to optimize the following min-max framework:

$$\arg\min_{\theta} \mathbb{E}_{(x,y)\sim\mathbb{D}} \left[ \max_{\epsilon\in\mathbb{S}} L(\theta, x + \epsilon, y) \right], \tag{1}$$

where $\mathbb{D}$ is the underlying data distribution, $L(\cdot,\cdot,\cdot)$ is the loss function, $\theta$ is the network parameter, $x$ is a training sample with the ground-truth label $y$, $\epsilon$ is the added adversarial perturbation, and $\mathbb{S}$ is the allowed perturbation range. Following [51, 48], the adversarial training here applies *single-step PGD* (PGD-1) to generate adversarial examples (for lowering training cost), with the constrain that maximum per-pixel change $\epsilon = 4/255$.

**Adversarial Training on Transformers.** We apply the setup above to adversarially train both ResNet-50 and DeiT-S. However, surprisingly, this default setup works for ResNet-50 but will collapse the training with DeiT-S, *i.e.*, the robustness of such trained DeiT-S is merely ~4% when evaluating against PGD-5. We identify the issue is over-regularization—when combining strong data augmentation strategies (*i.e.*, RangAug, Mixup and CutMix) with adversarial attacks, the yielded training samples are too hard to be learnt by DeiT-S.

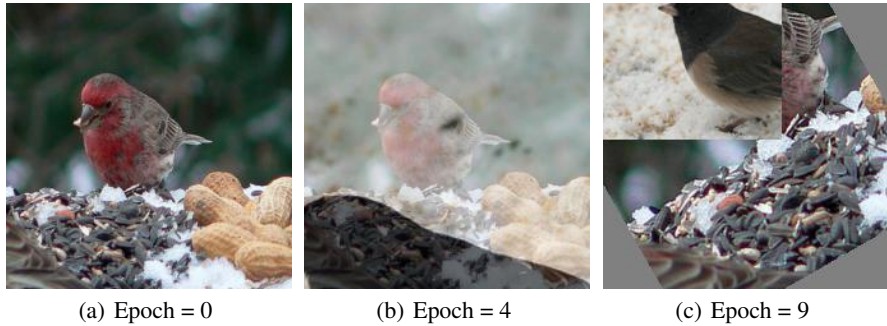

|  (a) Epoch = 0  |  (b) Epoch = 4  |  (c) Epoch = 9  |

Figure 1: The illustration of the proposed augmentation warm-up strategy. At the beginning of adversarial training (from epoch=0 to epoch=9), we progressively increase the augmentation strength.

To ease this observed training difficulty, we design a curriculum of the applied augmentation strategies. Specifically, as shown in Figure 1, at the first 10 epoch, we progressively enhance the augmentation strength (*e.g.*, gradually changing the distortion magnitudes in RandAug from 1 to 9) to warm-up the training process. Our experiment verifies this curriculum enables a successful adversarial training—DeiT-S now attains ~44% robustness (boosted from ~4%) on defending against PGD-5.

**Transformers with CNNs' Training Recipes.** Interestingly, an alternative way to address the observed training difficulty is directly adopting CNN's recipes to train Transformers [37], *i.e.*, applying M-SGD with step decay learning rate scheduler and removing strong data augmentation strategies (like Mixup). Though this setup can stabilize the adversarial training process, it significantly hurts the overall performance of DeiT-S—the clean accuracy drops to 59.9% (**-6.6%**), and the robustness on defending against PGD-100 drops to 31.9% (**-8.4%**).

One reason for this degenerated performance is that strong data augmentation strategies are not included in CNNs' recipes, therefore Transformers will be easily overfitted during training [6]. Another key factor here is the incompatibility between the SGD optimizer and Transformers. As explained in [25], compared to SGD, adaptive optimizers (like AdamW) are capable of assigning different learning rates to different parameters, resulting in consistent update magnitudes even with unbalanced gradients. This property is crucial for enabling successful training of Transformers, given the gradients of attention modules are highly unbalanced.

**CNNs with Transformers' Training Recipes.** As shown in Table 2, adversarially trained ResNet-50 is less robust than adversarially trained DeiT-S, *i.e.*, 32.26% *vs.* 40.32% on defending against PGD-100. It motivates us to explore whether adopting Transformers' training recipes to CNNs can enhance CNNs' adversarial training. Interestingly, if we directly apply AdamW to ResNet-50, the adversarial training will collapses. We also explore the possibility of adversarially training ResNet-50 with strong data augmentation strategies (*i.e.*, RandAug, Mixup and CutMix). However, we find ResNet-50 will be overly regularized in adversarial training, leading to very unstable training process, sometimes may even collapse completely.

Though Transformers' optimizer and augmentation strategies cannot improve CNNs' adversarial training, we find *Transformers' choice of activation functions matters*. Unlike the widely-used activation function in CNNs is ReLU, Transformers by default use GELU [18]. As suggested in [49], ReLU significantly weakens adversarial training due to its non-smooth nature; replacing ReLU with its smooth approximations (*e.g.*, GELU, SoftPlus) can strengthen adversarial training. We verify that by replacing ReLU with Transformers' activation function (*i.e.*, GELU) in ResNet-50. As shown in Table 2, adversarial training now can be significantly enhanced, *i.e.*, ResNet-50 + GELU substantially outperforms its ReLU counterpart by 8.01% on defending against PGD-100. Moreover, we note the usage of GELU enables ResNet-50 to match DeiT-S in adversarial robustness, *i.e.*, 40.27% *vs.* 40.32% for defending against PGD-100, and 35.51% *vs.* 35.50% for defending against AutoAttack, *challenging the previous conclusions [4, 37] that Transformers are more robust than CNNs on defending against adversarial attacks*.

Table 2: The performance of ResNet-50 and DeiT-S on defending against adversarial attacks (with $\epsilon = 4$). After replacing ReLU with DeiT's activation function GELU in ResNet-50, its robustness can match the robustness of DeiT-S.

|  | Activation | Clean Acc | PGD-5 | PGD-10 | PGD-50 | PGD-100 | AutoAttack |
|---|---|---|---|---|---|---|---|
| ResNet-50 | ReLU | 66.77 | 38.70 | 34.19 | 32.47 | 32.26 | 26.41 |
|  | GELU | 67.38 | 44.01 | 40.98 | 40.28 | 40.27 | 35.51 |
| DeiT-S | GELU | 66.50 | 43.95 | 41.03 | 40.34 | 40.32 | 35.50 |

## 4.2 Robustness to Patch-Based Attacks

We next study the robustness of CNNs and Transformers on defending against patch-based attacks. We choose Texture Patch Attack (TPA) [53] as the attacker. Note that different from typical patch-based attacks which apply monochrome patches, TPA additionally optimizes the pattern of the patches to enhance attack strength. By default, we set the number of attacking patches to 4, limit the largest manipulated area to 10% of the whole image area, and set the attack mode as the non-targeted attack. For ResNet-50 and DeiT-S, we do not consider adversarial training here as their vanilla counterparts already demonstrate non-trivial performance on defending against TPA.

Table 3: Performance of ResNet-50 and DeiT-S on defending against Texture Patch Attack.

| Architecture | Clean Acc | Texture Patch Attack |
|---|---|---|
| ResNet-50 | 76.9 | 19.7 |
| DeiT-S | 76.8 | **47.7** |

Interestingly, as shown in Table 3, though both models attain similar clean image accuracy, DeiT-S substantially outperforms ResNet-50 by 28% on defending against TPA. We conjecture such huge performance gap is originated from the differences in training setups; more specifically, it may be resulted by the fact DeiT-S by default use strong data augmentation strategies while ResNet-50 use none of them. The augmentation strategies like CutMix already naïvely introduce occlusion or image/patch mixing during training, therefore are potentially helpful for securing model robustness against patch-based adversarial attacks.

To verify the hypothesis above, we next ablate how strong augmentation strategies in DeiT-S (*i.e.*, RandAug, Mixup and CutMix) affect ResNet-50's robustness. We report the results in Table 4. Firstly, we note all augmentation strategies can help ResNet-50 achieve stronger TPA robustness, with improvements ranging from +4.6% to +32.7%. Among all these augmentation strategies, CutMix stands as the most effective one to secure model's TPA robustness, *i.e.*, CutMix alone can improve TPA robustness by 29.4%. Our best model is obtained by using both CutMix and RandAug, reporting 52.4% TPA robustness, which is even stronger than DeiT-S (47.7% TPA robustness). This observation still holds by using stronger TPA with 10 patches (increased from 4), *i.e.*, ResNet-50 now attains 34.5% TPA robustness, outperforming DeiT-S by 5.6%. *These results suggest that Transformers are also no more robust than CNNs on defending against patch-based adversarial attacks.*

Table 4: Performance of ResNet-50 trained with different augmentation strategies on defending against Texture Patch Attack. We note 1) all augmentation strategies can improve model robustness, and 2) CutMix is the most effective augmentation strategy to secure model robustness.

| Augmentations | | | Clean Acc | Texture Patch Attack |
|---|---|---|---|---|
| RandAug | MixUp | CutMix | | |
| ✗ | ✗ | ✗ | 76.9 | 19.7 |
| ✓ | ✗ | ✗ | 77.5 | 24.3 (+4.6) |
| ✗ | ✓ | ✗ | 75.9 | 31.5 (+11.8) |
| ✗ | ✗ | ✓ | 77.2 | 49.1 (+29.4) |
| ✓ | ✓ | ✗ | 75.7 | 31.7 (+12.0) |
| ✓ | ✗ | ✓ | 76.7 | **52.4 (+32.7)** |
| ✗ | ✓ | ✓ | 77.1 | 39.8 (+20.1) |
| ✓ | ✓ | ✓ | 76.4 | 48.6 (+28.9) |

## 5   Robustness on Out-of-distribution Samples

In addition to adversarial robustness, we are also interested in comparing the robustness of CNNs and Transformers on out-of-distribution samples. We hereby select three datasets, *i.e.*, ImageNet-A, ImageNet-C and Stylized ImageNet, to capture the different aspects of out-of-distribution robustness.

### 5.1   Aligning Training Recipes

We first provide a direct comparison between ResNet-50 and DeiT-S with their default training setup. As shown in Table 5, we observe that, *even without pretraining on (external) large scale datasets*, DeiT-S still significantly outperforms ResNet-50 on ImageNet-A (+9.0%), ImageNet-C (+9.9) and Stylized-ImageNet (+4.7%). It is possible that such performance gap is caused by the differences in training recipes (similar to the situation we observed in Section 4), which we plan to ablate next.

Table 5: DeiT-S shows stronger robustness generalization than ResNet-50 on ImageNet-C, ImageNet-A and Stylized-ImageNet. Note the results on ImageNet-C is measured by mCE (lower is better).

| Architecture | ImageNet ↑ | ImageNet-A ↑ | ImageNet-C ↓ | Stylized-ImageNet ↑ |
|---|---|---|---|---|
| ResNet-50 | 76.9 | 3.2 | 57.9 | 8.3 |
| ResNet-50* | 76.3 | 4.5 | 55.6 | 8.2 |
| DeiT-S | 76.8 | **12.2** | **48.0** | **13.0** |

**A fully aligned version.** A simple baseline here is that we completely adopt the recipes of DeiT-S to train ResNet-50, denoted as ResNet-50*. Specifically, this ResNet-50* will be trained with AdamW optimizer, cosine learning rate scheduler and strong data augmentation strategies. Nonetheless, as reported in Table 5, ResNet-50* only marginally improves ResNet-50 on ImageNet-A (+1.3%) and ImageNet-C (+2.3), which is still much worse than DeiT-S on robustness generalization.

It is possible that completely adopting the recipes of DeiT-S overly regularizes the training of ResNet-50, leading to suboptimal performance. To this end, we next seek to discover the "best" setups to train ResNet-50, by ablating learning rate scheduler (step decay *vs.* cosine decay), optimizer (M-SGD *vs.* AdamW) and augmentation strategies (RandAug, Mixup and CutMix) progressively.

**Step 1: aligning learning rate scheduler.** It is known that switching learning rate scheduler from step decay to cosine decay improves model accuracy on clean images [2]. We additionally verify that such trained ResNet-50 (second row in Table 6) attains slightly better performance on ImageNet-A (+0.1%), ImageNet-C (+1.0) and Stylized-ImageNet (+0.1%). Given the improvements here, we will use cosine decay by default for later ResNet training.

**Step 2: aligning optimizer.** We next ablate the effects of optimizers. As shown in the third row in Table 6, switching optimizer from M-SGD to AdamW weakens ResNet training, *i.e.*, it not only decreases ResNet-50's accuracy on ImageNet (-1.0%), but also hurts ResNet-50's robustness generalization on ImageNet-A (-0.2%), ImageNet-C (-2.4) and Stylized-ImageNet (-0.3%). Given this degenerated performance, we stick to M-SGD for later ResNet-training.

Table 6: The robustness generalization of ResNet-50 trained with different learning rate schedulers and optimizers. Nonetheless, compared to DeiT-S, all the resulted ResNet-50 show worse generalization on out-of-distribution samples.

|  | Optimizer-LR Scheduler | ImageNet ↑ | ImageNet-A ↑ | ImageNet-C ↓ | Stylized-ImageNet ↑ |
|---|---|---|---|---|---|
| | SGD-Step | 76.9 | 3.2 | 57.9 | 8.3 |
| ResNet-50 | SGD-Cosine | 77.4 | 3.3 | 56.9 | 8.4 |
| | AdamW-Cosine | 76.4 | 3.1 | 59.3 | 8.1 |
| DeiT-S | AdamW-Cosine | 76.8 | **12.2** | **48.0** | **13.0** |

**Step 3: aligning augmentation strategies.** Compared to ResNet-50, DeiT-S additionally applied RandAug, Mixup and CutMix to augment training data. We hereby examine whether these augmentation strategies affect robustness generalization. The performance of ResNet-50 trained with different combinations of augmentation strategies is reported in Table 7. Compared to the vanilla counterpart, nearly all the combinations of augmentation strategies can improve ResNet-50's generalization on out-of-distribution samples. The best performance is achieved by using RandAug + Mixup, outperforming the vanilla ResNet-50 by 3.0% on ImageNet-A, 4.6 on ImageNet-C and 2.4% on Stylized-ImageNet.

Table 7: The robustness generalization of ResNet-50 trained with different combinations of augmentation strategies. We note applying RandAug + Mixup yields the best ResNet-50 on out-of-distribution samples; nonetheless, DeiT-S still significantly outperforms such trained ResNet-50.

| Architecture | Augmentation Strategies | | | ImageNet ↑ | ImageNet-A ↑ | ImageNet-C ↓ | Stylized-ImageNet ↑ |
|---|---|---|---|---|---|---|---|
| | RandAug | MixUp | CutMix | | | | |
| | ✗ | ✗ | ✗ | 77.4 | 3.3 | 56.9 | 8.4 |
| | ✓ | ✓ | ✗ | 75.7 | **6.3** | **52.3** | **10.8** |
| ResNet-50 | ✓ | ✗ | ✓ | 76.7 | 6.3 | 56.3 | 7.1 |
| | ✗ | ✓ | ✓ | 77.1 | 6.1 | 55.1 | 8.8 |
| | ✓ | ✓ | ✓ | 76.4 | 5.5 | 54.0 | 9.1 |
| DeiT-S | ✓ | ✓ | ✓ | **76.8** | **12.2** | **48.0** | **13.0** |

**Comparing ResNet with the "best" training recipes to DeiT-S.** With the ablations above, we can conclude that the "best" training recipes for ResNet-50 (denoted as ResNet-50-Best) is by applying M-SGD optimizer, scheduling learning rate using cosine decay, and augmenting training data using RandAug and Mixup. As shown in the second row of Table 7, ResNet-50-Best attains 6.3% accuracy on ImageNet-A, 52.3 mCE on ImageNet-C and 10.8% accuracy on Stylized-ImageNet.

Nonetheless, interestingly, we note DeiT-S still shows much stronger robustness generalization on out-of-distribution samples than our "best" ResNet-50, *i.e.*, +5.9% on ImageNet-A, +4.3 on ImageNet-C and +2.2% on Stylized-ImageNet. *These results suggest that the differences in training recipes (including the choice of optimizer, learning rate scheduler and augmentation strategies) is not the key for leading the observed huge performance gap between CNNs and Transformers on out-of-distribution samples.*

**Model size.** To further validate that Transformers are indeed more robust than CNNs on out-of-distribution samples, we hereby extend the comparisons above to other model sizes. Specifically, we consider the comparison at a smaller scale, *i.e.* ResNet-18 (~12 million parameters) *vs.* DeiT-Mini (~10 million parameters, with embedding dimension = 256 and number of head = 4). For ResNet training, we consider both the fully aligned recipe version (denoted as ResNet*) and the "best" recipe version (denoted as ResNet-Best). Figure 2 shows the main results. Similar to the comparison between ResNet-50 and DeiT-S, DeiT-Mini also demonstrates much stronger robustness generalization than ResNet-18* and ResNet-18-Best.

We next study DeiT and ResNet at a more challenging setting—comparing DeiT to a much larger ResNet on robustness generalization. Surprisingly, we note in both cases, DeiT-Mini *vs.* ResNet-50 and DeiT-S *vs.* ResNet-101, DeiTs are able to show similar, sometimes even superior, performance than ResNets. For example, DeiT-S beats the nearly 2× larger ResNet-101* (~22 million parameters *vs.* ~45 million parameters) by 3.37% on ImageNet-A, 1.20 on ImageNet-C and 1.38% on Stylized-ImageNet. All these results further corroborate that Transformers are much more robust than CNNs on out-of-distribution samples.

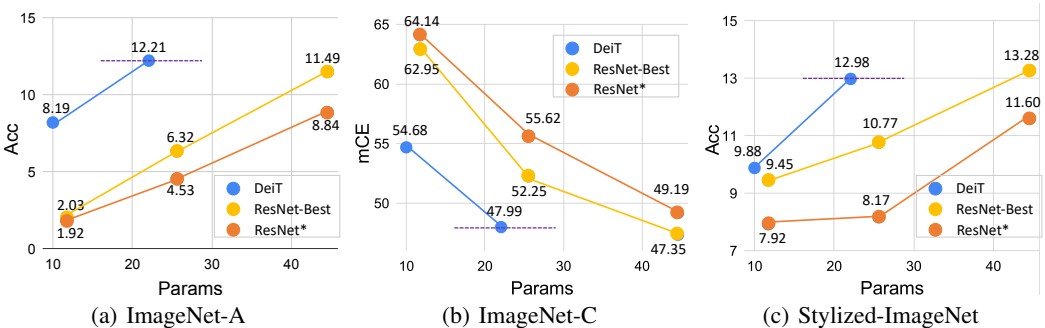

(a) ImageNet-A       (b) ImageNet-C       (c) Stylized-ImageNet

Figure 2: By comparing models at different scales, DeiT consistently outperforms ResNet* and ResNet-Best by a large margin on ImageNet-A, ImageNet-C and Stylized-ImageNet.

## 5.2 Distillation

In this section, we make another attempt to bridge the robustness generalization gap between CNNs and Transformers—we apply knowledge distillation to let ResNet-50 (student model) directly learn from DeiT-S (teacher model). Specifically, we perform soft distillation [20], which minimizes the Kullback-Leibler divergence between the softmax of the teacher model and the softmax of the student model; we adopt the training recipe of DeiT during distillation.

**Main results.** We report the distillation results in Table 8. Though both models attain similar clean image accuracy, the student model ResNet-50 shows much worse robustness generalization than the teacher model DeiT-S, *i.e.*, the performance is decreased by 7.0% on ImageNet-A, 6.2 on ImageNet-C and 3.2% on Stylized-ImageNet. This observation is counter-intuitive as student models typically achieve higher performance than teacher models in knowledge distillation.

However, interestingly, if we switch the roles of DeiT-S and ResNet-50, the student model DeiT-S is able to significantly outperforms the teacher model ResNet-50 on out-of-distribution samples. As shown in the third row and the fourth row in Table 8, the improvements are 6.4% on ImageNet-A, 6.3 on ImageNet-C and 3.7% on Stylized-ImageNet. *These results arguably suggest that the strong generalization robustness of DeiT is rooted in the architecture design of Transformer that cannot be transferred to ResNet via neither training setups or knowledge distillation.*

Table 8: The robustness generalization of ResNet-50, DeiT-S and their distilled models.

| Distillation | Architecture | ImageNet ↑ | ImageNet-A ↑ | ImageNet-C ↓ | Stylized-ImageNet ↑ |
|---|---|---|---|---|---|
| Teacher | DeiT-S | 76.8 | 12.2 | 48.0 | 13.0 |
| Student | ResNet-50*-Distill | 76.7 | 5.2 (-7.0) | 54.2 (+6.2) | 9.8 (-3.2) |
| Teacher | ResNet-50* | 76.3 | 4.5 | 55.6 | 8.2 |
| Student | DeiT-S-Distill | 76.2 | 10.9 (+6.4) | 49.3 (-6.3) | 11.9 (+3.7) |

## 5.3 Hybrid Architecture

Following the discussion in Section 5.2, we hereby ablate whether incorporating Transformer's self-attention-like architecture into model design can help robustness generalization. Specifically, we create a hybrid architecture (named Hybrid-DeiT) by directly feeding the output of res_4 block in ResNet-18 into DeiT-Mini, and compare its robustness generalization to ResNet-50 and DeiT-Small. Note that under this setting, these three models are at the same scale, *i.e.*, hybrid-DeiT (~21 million parameters) *vs*. ResNet-50 (~25 million parameters) *vs*. DeiT-S (~22 million parameters). We apply the recipe of DeiT to train these three models.

**Main results.** We report the robustness generalization of these three models in Figure 3. Interestingly, with the introduction of Transformer blocks, Hybrid-DeiT is able to achieve better robustness generalization than ResNet-50, *i.e.*, +1.1% on ImageNet-A and +2.5% on Stylized-ImageNet, *suggesting Transformer's self-attention-like architectures is essential for boosting performance on out-of-distribution samples*. We additionally compare this hybrid architecture to the pure Transformer architecture. As expected, Hybrid-DeiT attains lower robustness generalization than DeiT-S, as shown in Figure 3.

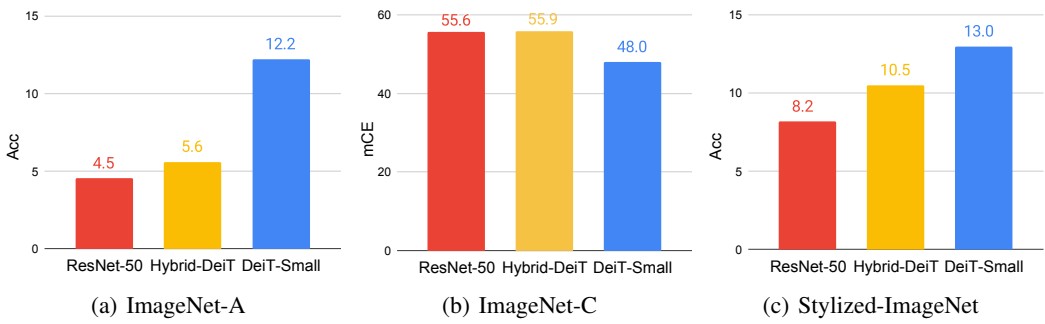

| (a) ImageNet-A | (b) ImageNet-C | (c) Stylized-ImageNet |

Figure 3: The robustness generalization of ResNet-50, DeiT-S and Hybrid-DeiT. We note introducing Transformer blocks into model design benefits generalization on out-of-distribution samples.

## 5.4 300-Epoch Training

As mentioned in Section 3.1, we by default train all models for only 100 epochs. This is a standard setup in training CNNs [15, 33], but not typical in training Transformers [44, 26]. To rule out the possibility of introducing negative effects in shortening training length, we lastly ablate the 300-epoch setup, *i.e.*, we directly borrow the default setup in [44] to train both ResNet and DeiT.

As reported in Table 9, DeiT-S substantially outperforms ResNet-50 by 10.4% on ImageNet-A, 7.5 on ImageNet-C and 5.6 on Stylized-ImageNet. Nonetheless, we argue that such comparison is less interesting and even unfair—DeiT-S already beats ResNet-50 by 1.8% on ImageNet classification, therefore it is expected that DeiT-S will also show stronger performance than ResNet-50 on ImageNet-A, ImageNet-C and Stylized-ImageNet.

Table 9: The robustness generalization of ResNet-50 and DeiT-S under the 300-epoch training setup. We note DeiT-S shows stronger performance than ResNet-50 on both clean images and out-of-distribution samples.

| Architecture | ImageNet ↑ | ImageNet-A ↑ | ImageNet-C ↓ | Stylized-ImageNet ↑ |
|---|---|---|---|---|
| ResNet-50 | 78.1 | 8.8 | 50.3 | 9.5 |
| DeiT-S | 79.9 | **19.2** | **42.8** | **15.1** |

To make the setup fairer (*i.e.*, comparing the robustness of models that have similar accuracy), we now compare DeiT-S to the much larger ResNet-101 (*i.e.*, ~22 million parameters *vs*. ~45 million parameters). The results are shown in Table 10. We observer that though both models achieve similar accuracy on ImageNet, DeiT-S demonstrates much stronger robustness generalization than ResNet-101. This observation can also holds for bigger Transformers and CNNs, *e.g.*, DeiT-B can consistently outperforms ResNet-200 on ImageNet-A, ImageNet-C and Stylized- ImageNet, despite they attain similar clean image accuracy (*i.e.*, 81.8% *vs*. 82.1%).

Table 10: The robustness generalization of ResNet and DeiT under the 300-epoch training setup. Though both models attain similar clean image accuracy, DeiTs show much stronger robustness generalization than ResNets.

| Architecture | ImageNet ↑ | ImageNet-A ↑ | ImageNet-C ↓ | Stylized-ImageNet ↑ |
|---|---|---|---|---|
| ResNet-101 | 80.2 | 17.6 | 45.8 | 11.9 |
| DeiT-S | 79.9 | **19.2** | **42.8** | **15.1** |
| ResNet-200 | 82.1 | 23.8 | 40.8 | 13.6 |
| DeiT-B | 81.8 | **27.9** | **38.0** | **17.9** |

In summary, in this 300-epoch training setup, we can draw the same conclusion as the one in the 100-epoch training setup, *i.e.*, *Transformers are truly much more robust than CNNs on out-of-distribution samples*. In addition, we note this conclusion is further corroborated in concurrent works [58, 30, 50, 62, 29], where a range of additional out-of-distribution tasks/datasets are tested. We refer interested readers to their papers for details.

## 6 Conclusion

With the recent success of Transformer in visual recognition, researchers begin to study its robustness compared with CNNs. While recent works suggest that Transformers are much more robust than CNNs, their comparisons are not fair in many aspects, *e.g.*, training datasets, model scales, training strategies, *etc*. This motivates us to provide a fair and in-depth comparisons between CNNs and Transformers, focusing on adversarial robustness and robustness on out-of-distribution samples. With our unified training setup, we found that Transformers are no more robust than CNNs on adversarial robustness. By properly adopting Transformer's training recipes, CNNs can achieve similar robustness as Transformers on defending against both perturbation-based adversarial attacks and patch-based adversarial attacks. While regarding generalization on out-of-distribution samples (*e.g.*, ImageNet-A, ImageNet-C and Stylized ImageNet), we find Transformer's self-attention-like architectures is the key. We hope this work would shed lights on the understanding of Transformer, and help the community to fairly compare robustness between Transformers and CNNs.

## Acknowledgements

This work was partially supported by the ONR N00014-20-1-2206, ONR N00014-18-1-2119 and Institute for Assured Autonomy at JHU with Grant IAA 80052272. Cihang Xie was supported by a gift grant from Open Philanthropy.

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
