# OpenReview forum: "Are Transformers more robust than CNNs? "
_NeurIPS.cc/2021/Conference — NeurIPS 2021 Poster_

### Official Review · Reviewer_HoyY · 2021-07-15

**Rating:** 7
**Confidence:** 3

**Summary:**

The authors compare the robustness to adversarial attacks and out-of-domain transfer of CNNs (ResNet-50) and Transformers (DeiT-S). First, they report that there is no difference between the two models in terms of robustness to adversarial attacks when dataset and data augmentation are comparable and training setup is optimized for each model such that the comparison is fair. Second, they report that Tranformers exhibit better generalization to out-of-domain samples than ResNets even under fair comparison, and attribute this difference to the self-attention architecture of the Transformer rather than differences in training protocol, datasets or augmentation strategies. Overall, it's a nice paper that tries to carefully compare two models that have been trained in very different ways in order to differentiate between effects of architecture and training protocol.

### Update after discussion period

I find the authors' response convincing and maintain my initial assessment to recommend acceptance.

**Limitations And Societal Impact:**

No limitations have been discussed. For instance, the authors could discuss the issues related to evaluating adversarial robustness and making a "fair" comparison between two models that require different training schemes to work well.

**Main Review:**

## Strengths
+ In-depth investigation how training protocol affects robustness
+ Reports that both architectures are equally robust/vulnerable to adversarial attacks
+ Provides evidence that Transformer architecture is more robust to out-of-domain samples

## Weaknesses
- Feels somewhat anecdotal/not very systematic at places
- Not sure how reliable the adversarial robustness comparison is


## Detailed comments

- It seems like the authors had to do quite some tweaking to improve adversarial training on DieT. If such tweaks lead to 6–8% differences in accuracy, I wonder how reliable and comparable the numbers in Table 2 really are. Wouldn't we expect further tweaking of one of the two models' training procedure to potentially change the picture completely? How would one decide which amount of tweaking to each of the models is "fair"?

- The fact that the numbers in Table 2 are *exactly* the same sounds like a bug in the evaluation script. It's really extremely unlikely to happen, so I urge the authors to double-check their code extra carefully.

- Why do the authors perform the investigation of different augmentation strategies only for the patch attack, but not for the attacks shown in Table 2? Lines 179f. seem to argue that the same augmentation as on DieT would harm ResNet's clean accuracy, but it's not clear to me why that would be the case if it has very little effect on clearn accuracy in Table 3.

- The motivation of studying the PatchAttack is not quite clear to me. It appears to be a much weaker attack for these two models, so what do we learn from the fact that data augmentation is effective against it for both models? Since the difficulty in testing adversarial robustness lies in finding adversarials, isn't only the most effective attack potentially indicative of robustness?

- Is training for the same number of epochs "fair"? The CNN has a much stronger inductive bias (shift equivariance) than the Transformer, so shouldn't we expect it to learn faster? I think it would make more sense to train both models to convergence rather than stopping at an arbitrary (convenient?) number of epochs.


**Time Spent Reviewing:**

3

---

> ### Author Response · Authors · 2021-08-10
> **Thanks for your appreciation & our robustness evaluation is reliable**
>
> We first thank the reviewer for the detailed comments and the appreciation of our work. We address the concerns below:
>
> **Q1: The effects of training setups on adversarial robustness?**
>
> A1: Thanks for raising this concern. First of all, we would like to highlight that ours is the first work that successfully applies adversarial training to Vision Transformer on ImageNet. The key (but also simple) technique is summarized in Fig 1---there should be a warm-up stage for data augmentation; otherwise, the whole adversarial training will collapse.
>
> Secondly, as analyzed in Smooth Adversarial Training [1], we believe tweaking other setups in adversarial training will not yield significant performance improvement. By comparing to the benefits brought by using smooth activation function (e.g., replace ReLU with GELU), [1] shows that other training enhancements (including change data augmentation, doubling training epoch) will only lead to a relatively small improvement (see Section 5.2 in [1], +0.6% for robustness). We refer the reviewer to [1] for more details.
>
> In summary, based on these prior works in adversarial training, we think tweaking other training parameters will not lead to significant robustness improvements and believe the presented adversarial robustness comparison is reliable. Nonetheless, we reserve our attitude about tweaking training setups for further improving adversarial robustness of Transformers (given this direction is under-explored and this work is the first one that successfully makes adversarial training works for Vision Transformers on ImageNet); we believe this part is non-trivial and leave it as future work.
>
> **Q2: Bugs in evaluation script?**
>
> A2: First of all, we would like to stress that we use the open-sourced & renowned AutoAttack package (https://github.com/fra31/auto-attack) for our robustness evaluations. This package is well written and easy to use---the only thing that needs to be done on our side is to provide model definition and weights; all other attack procedures will be automatically done by the package. We also re-evaluate the models and find the results are the same.
>
> To further alleviate the reviewer’s concern, we attach the evaluation script with checkpoints & logs in this anonymous github repo: https://anonymous.4open.science/r/evaluation-52CB. Please let us know if you see anything wrong/suspicious.
>
> **Q3: Data augmentation strategies on adversarial robustness?**
>
> A3: In line 179, the main reason that we observe performance degrade is that we only adversarially train models for 100 epochs. As argued in [1], given adversarial training is much harder than standard training, adding strong data augmentation will make the whole training more difficult, therefore needing a longer training schedule to cope with it. Nonetheless, as shown in [1] and answered in Q1, data augmentation still plays a relatively small role in improving adversarial robustness even if we increase the training epoch from 100 to 200.
>
> In Table 3, we first stress that we are performing standard training now rather than adversarial training. Surprisingly, we find data augmentation plays a vital role in improving robustness against patch attacks (i.e., cutmix can substantially enhance model robustness against patch attacks) while the impact from architecture design (e.g., choose CNN or Transformer) is relatively small.
>
> We will update the paper accordingly to make the training setting in Table 2&3 more clear.
>
> **Q4: Why study PatchAttack?**
>
> A4: Sorry for the confusion. The main motivation for us to specifically study PatchAttack is due to the evaluation suite design in AutoAttack. In AutoAttack, two types of adversarial attacks are considered: one is gradient-based PGD attacks, and another is gradient-free SquareAttack [2]. In prior studies, the analysis on these two attacks is mostly mixed together. Nonetheless, in this paper, to provide the general audience a better understanding of models' robustness against different attacks, we decide to provide an in-depth & separate analysis of PGD attacks (in Table 2) and SquareAttack. Regarding SquareAttack, we further enhance it by using PatchAttack for our analysis.
>
> Interestingly, we found the key to improve robustness against these two different types of adversarial attacks are different (one largely relies on architecture design, while another largely relies on data augmentation strategies). This observation also confirms the need of analyzing different attacks in a separate manner. We will clarify this motivation in the paper.
>
> **Q5: Concerns on model convergence?**
>
> A5: Thanks for raising this concern. First of all, we would like to clarify that 100 training epoch is quite standard for network training. For example, in this popular pytorch repo https://github.com/facebookresearch/pycls/tree/master/configs/dds_baselines, all models (including efficientnet, resnet) are trained with 100 epochs for fair performance comparison. Additionally, the usage of cosine learning rate (which is also used in this github repo) can guarantee models will converge to local optima. For example, we observe both CNNs’ training loss and Transformers’ training loss are flatten at the end of training.
>
> Secondly, we also follow the DeiT’s original setup by training both models for 300 epochs (please see our analysis in the general response to all reviewers). In short, under this new setting, we can still draw the conclusion that Transformer is much more robust CNNs on out-of-distribution samples.
>
> Lastly, we stress that by training models for the same number of epoch is a standard setting in many prior works [3] for fair comparisons. However, we do agree that there should also be other ways to construct fair comparisons. We believe it is a promising & interesting direction to compare CNNs & Transformers from different “fair” perspectives/criteria and decide to leave it as a future work.
>
>
>
>
> [1] Xie, C., Tan, M., Gong, B., Yuille, A., & Le, Q. V. (2020). Smooth adversarial training. arXiv preprint arXiv:2006.14536.
> [2] Andriushchenko, Maksym, et al. "Square attack: a query-efficient black-box adversarial attack via random search." European Conference on Computer Vision. Springer, Cham, 2020.
> [3] ​​Radosavovic, Ilija, et al. "Designing network design spaces." Proceedings of the IEEE/CVF Conference on Computer Vision and Pattern Recognition. 2020.

---

### Official Review · Reviewer_Nz2M · 2021-07-16

**Rating:** 6
**Confidence:** 3

**Summary:**

This paper compare the robustness of transformers and CNNs.
They identify some pitfalls of the evaluation protocol and redo the benchmark of adversarial robustness & OOD generalization.
The conclusion is interesting: 1. VIT is not really more robust than CNNs. 2. VIT indeed better generalize on OOD samples.

**Main Review:**

Strength:
1. They identify some pitfalls of comparing robustness between CNNs and transformers: e.g., number of parameters, dataset, data augmentation, training hyper-parameter.
2. In Table 2, it seems like GELU activation is main source of robustness of VIT.
3. The empirical study on OOD samples is comprehensive. It is a strong evidence showing that VIT indeed better generalize on OOD samples.

Weakness & Questions:
1. Is number of training epoch a big problem when evaluating the robustness. I think it should be fine as long as both models converge to the local optima.
2. As you mentioned in line 105, 100 epoches are relatively short for training resnet-50/vit. Is the comparison fair? Shouldn't both model be trained to converge and achieve the lowest training loss / best validation performance?
3. It would be also interesting to see how scaling would affect the robustness of VIT and CNNs (like Figure 2).

**Time Spent Reviewing:**

1

---

> ### Author Response · Authors · 2021-08-10
> **Thanks for your appreciation to our work**
>
> We first thank the reviewer for the detailed comments and the appreciation of our work. We address the concerns below:
>
> **Q1: Concern on the number of training epoch**
>
> A1: Thanks for raising this concern. First of all, we would like to clarify that 100 training epoch is quite standard for network training. For example, in this popular pytorch repo https://github.com/facebookresearch/pycls/tree/master/configs/dds_baselines, all models (including efficientnet, resnet) are trained with 100 epochs for fair performance comparison. Additionally, the usage of cosine learning rate (which is also used in this github repo) can guarantee models will converge to local optima. For example, we observe both CNNs’ training loss and Transformers’ training loss are flatten at the end of training.
>
> **Q2: 100 epoches are relatively short?**
>
> A2: As mentioned in Q1, 100 epoch is quite standard for network training and we do observe that both CNNs and Transformers converge by the end of training. But as requested, we additionally provide the analysis under the 300 epoch training setup (see our general response to all reviewers above).  In short, under such a setup, we can still draw the conclusion that Transformers are much more robust than CNNs on out-of-distribution samples.
>
> **Q3: How scaling affects the robustness of ViT and CNNs (like Figure 2).**
>
> A3: Thanks for your suggestion. As confirmed in many prior works on adversarial training [1], larger models will yield better adversarial robustness. We expect such a phenomenon will also be observed in ours as well. Nonetheless, given both scaling to larger models (e.g., ResNet-101, DeiT-B) and performing adversarial training significantly increase training cost, our computational resources are not enough for supporting these ablations during this short rebuttal period. Besides, given the “non-promising” signal that Transformers are no more robust than CNNs in adversarial robustness, we tend to not rate it as a top priority task for this paper.
>
> [1] Madry, Aleksander, et al. "Towards deep learning models resistant to adversarial attacks." arXiv preprint arXiv:1706.06083 (2017).

---

> > ### Comment · Reviewer_Nz2M · 2021-08-26
> > **Post Rebuttal Notes**
> >
> > It's good to see the new experiment results and discussion.
> > It would be great to see them in the next version of the paper.

---

### Official Review · Reviewer_rDdC · 2021-07-16

**Rating:** 6
**Confidence:** 3

**Summary:**

This paper investigates the robustness of vision transformers and compare them with CNNs. It show that transformers do not show superiority to CNNs from the perspective of robustness.  Extensive experiments are conducted to validate the conclusions.

**Limitations And Societal Impact:**

It has been addressed.

**Main Review:**

Strength:

- It is interesting that this paper challenges the conclusions of previous works.  The right principles will be obtained after such challenging and discussions.

- This paper seems to make a fairer comparison between CNNs and transformers by considering the training strategies. The robustness of transformers can be understood better.

Weakness:

- It seem that the training recipes have a large impact on the final results. It is interesting to see that how the training recipes (e.g., learning rate, epochs, optimizer) affect the robustness of both transformers and CNN？

- Could the observation of this paper inspires the occurrence of new method for improving the robustness? The authors are required to make more discussion.


**Time Spent Reviewing:**

4h

---

> ### Author Response · Authors · 2021-08-10
> **Thanks for your comments**
>
> We first thank the reviewer for the detailed comments and the appreciation of our work. We address the concerns below:
>
> **Q1: The impact of training recipes on robustness?**
>
> A1: Thanks for your comments. Firstly, in Section 5, we carefully ablate different training setups (e.g., learning rate, optimizer) and show how they affect models’ robustness generalization on out-of-distribution samples. We additionally provide the effect of training epoch on robustness generalization in this rebuttal (see our general response to all reviewers)
>
> Regarding such ablation for adversarial robustness, in Section 4.2.1, we analyze the effects of optimizers. Specifically, for DeiT-S, if we change the optimizer from AdamW to M-SGD, its accuracy will decrease by 6.6% & its PGD-100 robustness will decrease by 8.4%. As suggested by [1], we conjecture this phenomenon is mainly induced by the fact SGD is innately not suitable for optimizing Transformers. Due to time constraints and resource limitation, we do not have enough bandwidth to ablate other training recipes in this rebuttal, but we believe a similar phenomenon will be observed as in [2]: enhancing training recipes (including change data augmentation, doubling training epoch) will only lead to a relatively small improvement, i.e., +1.7% for accuracy and +0.6% for robustness (see Section 5.2 in [2]).
>
> **Q2: Inspiring new ideas/methods?**
>
> A2: Thanks for this great suggestion. We believe our analysis can spark the community’s interest in further exploring the potential of Transformer on robustness generalization. More concretely, our analysis here points out that self-attention-like architecture is the most essential key for obtaining strong robust generalization, so one potential direction to go is to further tweak the structural design of the Transformer block (e.g., properly combine the local attention (for texture information) & the long-range attention (for shape information)); alternatively, another promising direction to go is to provide deeper analysis on why Transformer is better at robustness generalization, e.g., explaining what key/unique information is learned by Transformer blocks for enabling this strong generalization ability.
>
> We will further expand this discussion and add it to our paper.
>
> [1] Liyuan Liu, Xiaodong Liu, Jianfeng Gao, Weizhu Chen, and Jiawei Han. Understanding the difficulty of training transformers. arXiv preprint arXiv:2004.08249, 2020.
>
> [2] Xie, C., Tan, M., Gong, B., Yuille, A., & Le, Q. V. (2020). Smooth adversarial training. arXiv preprint arXiv:2006.14536.

---

### Official Review · Reviewer_bh3x · 2021-07-18

**Rating:** 5
**Confidence:** 4

**Summary:**

The paper attempts to compare the robustness of Transformers and CNNs in fairer settings. The paper draws conclusions that Transformers are no better than CNNs in adversarial robustness, but are better at the generalization ability to out-of-distribution samples.

**Ethics Review Area:**

["I don’t know"]

**Limitations And Societal Impact:**

See above

**Main Review:**

Originality

-- the paper is mostly an experimental report. While it has values to the community, the new "fair" settings are not really fair enough. In fact, for such experimental/analysis paper, there is a higher requirements on the quality of experimental designs and implementations. I also did not see non-trivial or significant findings, or something surprising me.

Quality

-- the quality is borderline.

1) The ResNet-50 and DeiT-S are both not in their best/common status, e.g. DeiT-S should achieve as high as 79.xx% on ImageNet-1K, but the paper uses an implementation of 76.8%. It is not a must to have higher baseline, but since the augmentations/regularizations will also affect robustness a lot, it is better to be based on a correct implementation.
2) why GELU is better than ReLU? This is an interesting findings if solid proved, but the paper does not explore the reasons behind, or conduct more experiments to verify it, e.g. replacing GELU in DeiT-S by ReLU.
3) the authors argue that Transformers are better at generalization ability, and the ability comes from its structure (Line 277-279). The proof is not solid, actually, it needs more fine-grained comparison by varying only 1 component in the structure to draw some meaningful conclusions.

-- clarity

Clear.

-- significance

This is a right attempt, which will be valuable to the community. However, unfortunately, the experimental designs and implementations are not high-quality and solid enough to support meaningful conclusions.



**Time Spent Reviewing:**

4 hours

---

> ### Author Response · Authors · 2021-08-10
> **Our comparison is indeed correct & fair**
>
> We first thank the reviewer for the detailed comments and the acknowledgment that providing a “fair” comparison between Transformers and CNNs is a right and valuable research attempt to the community. We address the concerns below:
>
> **Q1: The implementation could be wrong, as DeiT-S should achieve as high as 79.xx% on ImageNet-1K.**
>
> A1: First of all, we would like to clarify that our codebase is exactly built upon the official DeiT github repo (https://github.com/facebookresearch/deit). The main reason that our DeiT-S only gets 76.8% top-1 accuracy is that we reduce the total training epoch number from 300 to 100 (which is ResNet’s default training epoch number) in ALL experiments. As detailed in the general response to all reviewers (see above), this strategy 1) can help us largely reduce the training budget; and 2) still leads to the similar observations/conclusion as the ones we draw from the 300 training epoch setup (where DeiT-S achieves 79.9% top-1 accuracy). We will update the paper accordingly for clarifying this training setup.
>
> &nbsp;
>
> **Q2: GELU v.s. ReLU?**
>
> A2: Regarding the ablation of GELU/ReLU in the ResNet architecture, it is already explained in the Smooth Adversarial Training [1] paper---by replacing a non-smooth activation function (e.g., ReLU) with a smooth activation function (e.g., GELU), it can help networks find harder adversarial examples and compute better gradient updates during adversarial training, therefore resulting in stronger adversarial robustness. We refer the reviewer to [1] for more details. In this paper, our results confirm the conclusion from [1], and found using GELU is the key to let ResNet achieve similar adversarial robustness as the Transformer.
>
> We additionally perform the ablation of GELU/ReLU in the Transformer architecture, and found similar results---Transformer+GELU achieves much higher adversarial robustness than Transformer+ReLU.
>
> [1] Xie, C., Tan, M., Gong, B., Yuille, A., & Le, Q. V. (2020). Smooth adversarial training. arXiv preprint arXiv:2006.14536.
>
> &nbsp;
>
> **Q3: Fine-grained comparison for justifying Transformer’s architecture benefits**
>
> A3: Besides using the distillation experiment in Section 5.4 to justify Transformer is a better architecture than CNN on robustness generalization, we also include hybrid architectures experiment in Section 5.5 to support this argument as well. Specifically, by restricting the parameter number to be similar with DeiT-S/ResNet-50, we properly combine ResNet blocks and Transformer blocks to construct a new hybrid architecture. As shown in Fig. 3, this hybrid architecture shows better performance than ResNet-50 on ImageNet-A (+1.1%) & Stylized-ImageNet (+2.5%), therefore can support that the self-attention-like architecture is the key for obtaining more robustness. We will update the paper accordingly to make this part more clear.
>
> &nbsp;
>
> In summary, we believe this paper provides a high-quality & fair robustness comparison between Transformers and CNNs. Besides, under our fair experiment setting, our paper draws significantly different conclusions from previous “unfair” explorations. We believe this study is important for the community to better understand the robustness potential of Transformer.

---

> > ### Comment · Reviewer_bh3x · 2021-09-02
> > **post rebuttal**
> >
> > The rebuttal partly addressed my questions, including Q1 and Q2. However, I am still concerned at the quality of experiments and insights, especially regarding this is an analysis paper which requires stronger experiments and deeper understanding than common algorithm paper. Hence, I maintain my original score.

---

> > > ### Author Response · Authors · 2021-09-02
> > > **Thanks for the comments.**
> > >
> > > Thanks for your comments.
> > >
> > > As for the experiment and evaluation, we are confident it is correct since
> > > 1) For training, we rely on a open-source codebase to build the experiments, (https://github.com/rwightman/pytorch-image-models), which is very stable and has been tested by many papers.
> > > 2) For robustness evaluation, we also rely on the open-sourced & renowned AutoAttack package (https://github.com/fra31/auto-attack).
> > >
> > > For your convenience, we have attached the evaluation script with checkpoints & logs in this anonymous github repo: https://anonymous.4open.science/r/evaluation-52CB, and will release our whole codebase as soon as possible.

---

### Author Response · Authors · 2021-08-10
**A General Response to the Concern on “Fair” Robustness Comparison (especially on the number of training epochs)**

We thank all reviewers for their thoughtful feedback, which will help us improve the quality of this paper. We are delighted to see all reviewers agree that our study on fair robustness comparison between Transformers and CNNs is interesting and valuable to the community, and appreciate the analysis presented in this paper. The major concern (suggested by all reviewers) is that our “fair” robustness comparison will be “fairer” if we could train both Transformers and CNNs for a longer epoch. In this general response, we would like to first clarify why our original experiment setup is already standard & fair, and then we will report the results of applying a longer training epoch on Transformers and CNNs.

&nbsp;

### Part 1: our original experiment setup is already standard & fair

To fairly compare the robustness between Transformers and CNNs, our work carefully aligns & ablates their training setups (e.g., data augmentation strategies, optimizers). Specifically, regarding the selection of training epochs, we have two choices: following either ResNet’s default setup (100 epoch, e.g., see the popular ResNet pytorch repo https://bit.ly/3fLqOSc) or Transformer’s default setup (300 epoch, e.g., see the official DeiT pytorch repo https://bit.ly/2VHf38a). Our paper finally chooses to follow ResNet’s standard (by training both models for 100 epochs), based on the following considerations:

1. The most representative architecture of ResNet (ResNet-50) and the most representative architecture of Transformer (DeiT-Small) is more comparable under this setting: they have a similar number of parameters (25M v.s. 22M) and a similar top-1 accuracy on ImageNet classification (76.3% v.s. 76.8%). The studied problem then becomes much more appealing: *for two neural architectures that perform similarly on ImageNet classification, will they exhibit differently on robustness evaluations?*

2. More practically, compared to the 300 epoch setup, 100 epoch setup significantly reduces the computational budget. This strategy not only enables us to have extra bandwidth to conduct in-depth robustness comparisons between Transformers and CNNs, but also make the future follow-ups easier to reproduce our results and to keep exploring this direction in a computationally efficient manner. More importantly, as we argued next, reducing the training epoch from 300 to 100 will not alter our conclusions.

We will make the justification of the training setup much more clear in the next version.

&nbsp;

### Part 2: 300 epoch training results

We then follow the reviewers’ comments by training both models for 300 epochs. Under this training setup, the results are as the following (%):

| Model      | ImageNet &uarr; | ImageNet-A &uarr; | ImageNet-C &darr; | Stylized-ImageNet &uarr;|
|------------|----------|------------|------------|-------------------|
| ResNet-50  | 78.1     | 8.8        | 50.3       | 9.5               |
| DeiT-Small | **79.9**     | **19.2**       | **42.8**       | **15.1**              |

From the table above, though we can still observe that the Transformer shows much stronger generalization than the CNN on out-of-distribution samples, *we argue that such setup is less interesting and a little bit unfair*---DeiT-Small shows much better accuracy than ResNet-50 on ImageNet classification, therefore it is expected that DeiT-Small will also show stronger robustness generalization than ResNet-50 on out-of-distribution samples.

To make the setup fairer (i.e., comparing the robustness of models that have similar accuracy), we now compare the robustness generalization between DeiT-Small and ResNet-101. The results are as the following (%):

| Model      | ImageNet &uarr; | ImageNet-A &uarr; | ImageNet-C &darr; | Stylized-ImageNet &uarr;|
|------------|----------|------------|------------|-------------------|
| ResNet-101 | **80.2**     | 17.6       |     45.8       | 11.9              |
| DeiT-Small | 79.9     | **19.2**       | **42.8**       | **15.1**              |

We can observe that though ResNet-101 and DeiT-Small achieve similar accuracy on ImageNet, DeiT-Small demonstrates much stronger robustness generalization than ResNet-101.

We further scale this setup by comparing DeiT-Base and ResNet-152. The results are shown below (%):


| Model      | ImageNet &uarr; | ImageNet-A &uarr; | ImageNet-C &darr; | Stylized-ImageNet &uarr;|
|------------|----------|------------|------------|-------------------|
| ResNet-152 | 81.2     | 21.3       |     42.1       | 12.3              |
| DeiT-Base | **81.8**   | **27.9**       | **38.0**       | **17.9**              |


In summary, in this 300 training epoch setup, we can still draw exactly the same conclusion that we presented in the paper (using 100 training epoch setup)---Transformers are much more robust than CNNs on out-of-distribution samples. We will add this 300 epoch training setup as an ablation in the next version.

---

### Decision · Program_Chairs · 2021-09-27

**Decision:**

Accept (Poster)

**Comment:**

This paper did a comparative study of the robustness between CNNs and Vision Transformers (ViTs). They found that after carefully adjusting training setups for fair comparisons, there is no difference in the robustness to adversarial attack between the two families of architectures. Meanwhile, ViTs generalize better on OOD samples, which the paper suggested was due to the self-attention architectures rather than training setups. The reviewers generally found those results interesting and valuable to the community. There were concerns about the evaluation setup and sub-par baseline results. The author responded in the rebuttal that this is mainly due to lower number of training epochs (100), and reported results with 300 training epochs.